# The Influence of Synthesis Parameters on Structural and Magnetic Properties of Iron Oxide Nanomaterials

**DOI:** 10.3390/nano10010085

**Published:** 2020-01-02

**Authors:** Laura Madalina Cursaru, Roxana Mioara Piticescu, Dumitru Valentin Dragut, Ioan Albert Tudor, Victor Kuncser, Nicusor Iacob, Florentin Stoiciu

**Affiliations:** 1National R&D Institute for Non-Ferrous and Rare Metals, INCDMNR-IMNR, 102 Biruintei blvd, Pantelimon, 077145 Ilfov, Romania; dragutv@imnr.ro (D.V.D.); atudor@imnr.ro (I.A.T.); fstoiciu@imnr.ro (F.S.); 2National Institute of Materials Physics, Atomistilor 105bis, P.O. Box MG-7, 077125 Bucharest-Magurele, Romania; kuncser@infim.ro (V.K.); nicusor.iacob@infim.ro (N.I.)

**Keywords:** iron oxide nanoparticles, hydrothermal synthesis, high pressure, magnetic properties, thermal stability

## Abstract

Magnetic iron oxides have been used in biomedical applications, such as contrast agents for magnetic resonance imaging, carriers for controlled drug delivery and immunoassays, or magnetic hyperthermia for the past 40 years. Our aim is to investigate the effect of pressure and temperature on the structural, thermal, and magnetic properties of iron oxides prepared by hydrothermal synthesis at temperatures of 100–200 °C and pressures of 20–1000 bar. It has been found that pressure influences the type of iron oxide crystalline phase. Thus, the results obtained by Mössbauer characterization are in excellent agreement with X-ray diffraction and optical microscopy characterization, showing that, for lower pressure values (<100 bar), hematite is formed, while, at pressures >100 bar, the major crystalline phase is goethite. In addition, thermal analysis results are consistent with particle size analysis by X-ray diffraction, confirming the crystallization of the synthesized iron oxides. One order of magnitude higher magnetization has been obtained for sample synthesized at 1000 bar. The same sample provides after annealing treatment, the highest amount of good quality magnetite leading to a magnetization at saturation of 30 emu/g and a coercive field of 1000 Oe at 10 K and 450 Oe at 300 K, convenient for various applications.

## 1. Introduction

Magnetic iron oxide particles have been used for in vitro diagnostics for the past 40 years. Due to the unique physical, chemical, thermal and mechanical properties of iron oxide nanoparticles, as well as their biocompatibility and low toxicity in the human body, they have been used in many biomedical applications [1,2,3,4,5,6], such as contrast agents [7] for magnetic resonance imaging (MRI), carriers for controlled drug delivery and immunoassays [8,9,10,11,12,13], and also in magnetic hyperthermia [14,15,16,17,18,19,20,21,22,23]. All these applications require the particles to be superparamagnetic at room temperature. Aggregation of nanoparticles should be avoided to prevent blockage of blood vessels. In addition, the particles’ stability in water at neutral pH is very important for these applications and the colloidal stability of the magnetic fluid depends on the coating materials and the particle size [24,25]. Magnetic iron oxide nanoparticles with polymer coatings have also been used in cell separation, protein purification and organic or biochemical syntheses [26,27,28,29,30]. Coatings are not only used to enhance stability, but also particle functionality. On the other hand, it is known that when the size of the magnetic material is close to or smaller than the size of the characteristic parameters, the effect of small particle sizes can influence the important physical properties of nanostructures, such as structural phase transition temperature or Néel temperature (the temperature at which an anti-ferromagnetic material becomes paramagnetic). For example, Jiang’s group found that changing the size and grain size of metal and alloy can alter the phase transition temperature [31].

Ruan et al. [31] studied the thermal stability of nano-rings and nanotubes of α-Fe_2_O_3_ and found that it is closely related to the surface fraction of (001) plane in nanostructures. It has also been found that the different thickness of α-Fe_2_O_3_ nano-rings may affect their thermal stability. Recently, numerous experimental studies have been conducted to investigate the size effect on phase transitions [31,32,33,34,35,36,37,38]. The stability of hematite (α-Fe_2_O_3_) of various morphologies has also been studied.

Most papers focused on oxidation from Fe_3_O_4_ to α-Fe_2_O_3_ or γ-Fe_2_O_3_ to α-Fe_2_O_3_, as these phase changes are found to be associated with particle size, nanostructure, and other factors. The quasi-cubic phase stability of α-Fe_2_O_3_ was studied using magnetic measurements at high temperatures in high vacuum. The results showed that phase transformation from α-Fe_2_O_3_ (low ferromagnetic hematite) to Fe_3_O_4_ (ferrimagnetic magnetite) strongly depends on the size of the structures. Fe_2_O_3_ presents a defined chemical phase (Fe^3+^) with high chemical stability, while the mixed chemical state of Fe_3_O_4_ (Fe^2+^/^3+^) could induce instability [39].

The spectroscopic and morphological characterization of Fe_2_O_3_ nanowires (NWs) was performed according to calcination temperature to assess the thermal stability of NWs and temperatures over which a chemical reduction of Fe ions occurred. Thermogravimetric measurements clearly show the reduction in mass due to oxygen loss, while infrared and photoemission measurements allow tracking of the reduction of iron ions at different temperatures, indicating chemical reduction to Fe_3_O_4_ starting at moderate temperatures (above 440 K).

According to literature data, magnetite nanomaterials are very vulnerable to oxidation in air at temperatures above 150 °C. During heating in the presence of oxygen, magnetite nanoparticles are transformed into maghemite and then into hematite. A similar effect of magnetite oxidation can be induced by laser irradiation [40,41]. At a macroscopic level, oxidation of magnetite to hematite at room temperature is inhibited and changes in the crystalline structure can be achieved only by heating at 600 °C. At the nanometric level, changes in crystalline structure can be expected and observed at much lower temperatures even close to room temperature. This is due to enthalpy and activation energy, which depend on particle size.

It was found that Fe nanoparticles are oxidized to a mixture of iron oxides (γ-Fe_2_O_3_ and α-Fe_2_O_3_) even at 200 °C. However, this temperature may vary due to the large surface area and the various activities of the nanoparticles, which leads to a higher exothermic process during low temperature oxidation. In general, it can be assumed that phase transformations in nano-granular systems occur from 200 °C to 600 °C with different contributions of oxides, γ-Fe_2_O_3_, and α-Fe_2_O_3_. It should also be emphasized that data on the behavior of nanosystems in high-temperature conditions are very different and generalizations cannot be made [42].

To our knowledge, there have been no investigations regarding the effect of high pressure on the formation and phase transformations of iron oxides in aqueous environments under hydrothermal conditions. In our previous work [43], the influence of the main synthesis parameters (temperature, time, pressure) on the formation of nanosized Fe_2_O_3_ particles has been studied up to the pressure of 20 bar.

In the present study, our aim is to investigate the effect of pressure (up to 1000 bar) and temperature, as main hydrothermal synthesis parameters on the formation of different crystalline phases of nanostructured iron oxides and on structural, thermal, and magnetic properties of iron oxide nanomaterials.

## 2. Materials and Methods 

### 2.1. Hydrothermal Synthesis of Hematite

FeCl_3_·6H_2_O, p.a, 99% (Merck KGaA, Darmstadt, Germany), and ammonia solution (Chimreactiv SRL, Bucharest, Romania) were used for hydrothermal synthesis of iron oxides.

In a first step, FeCl_3_·6H_2_O was dissolved in distilled water to obtain a solution whose iron concentration was determined by the Inductively coupled Plasma—Optical Emission Spectroscopy (ICP-OES) method. For the precipitation of the iron oxide precursors, a 25% ammonia solution was added dropwise under magnetic stirring. A brown precipitate with alkaline pH was obtained. Thus, obtained suspension was washed with water to remove the by-products.

In the second step, the washed suspension (iron oxide precursor) was transferred to the autoclave and subjected to hydrothermal synthesis at 200 °C for 3 h. SAM autoclave (Romania) endorsed with cooling system, was used to prepare samples at pressures of 20–100 bar (experimental conditions: working volume 0.3 L; pressure created inside the stainless steel vessel of the autoclave using argon gas; temperature 200 °C), while HP Systems autoclave (Bordeaux, France) was used in the case of sample obtained at 1000 bar (experimental conditions: working volume 1 L; isostatic pressure; temperature 100 °C). After hydrothermal treatment, the nanostructured powders were dried by lyophilization using a Martin Christ Alpha 1–2 LD Plus freeze dryer (City, US State abbrev. if applicable, Country). Experimental parameters of the investigated samples and chemical analysis results are presented in Table 1. Fe content was determined using a chemical quantitative method, according to STAS 1574/3-90.

### 2.2. Characterization Methods

The crystal structure and the phases present in the synthesized nanopowders were analyzed using X-ray phase analysis. X-ray diffraction (XRD) patterns were obtained by Bruker-AXS D8 ADVANCE diffractometer (Bruker AXS GmbH, Karlsruhe, Germany) equipped with CuKα radiation source and a scintillation detector with graphite monochromator in vertical geometry θ-θ. XRD patterns were recorded in the range of 2θ = 14–84° at the scan speed of 0.02 degrees/s, using DIFFRAC.EVA version 2016 software (Bruker AXS GmbH, Karlsruhe, Germany) and the ICDD PDF 4+ 2019 database. The chemical structure of the synthesized products was determined by Fourier Transform Infrared spectroscopy (FT-IR) within the scanning range of 550–4000 cm^−1^ in transmittance mode. Measurements were performed using a FT-IR ABB MB 3000 spectrometer (ABB Inc., Québec, QC, Canada) equipped with a Diffuse reflection accessory, EasiDiff device (PIKE Technologies, Inc., Madison, WI, USA) for qualitative powder analysis. The solid sample was mixed with KBr powder so that its concentration in the mixture was 1 wt%. The mixture thus obtained is milled for 15 min to grind grains and obtain fine, homogeneous particles. For data acquisition, 64 scans were made at an optical resolution of 4 cm^−1^. Experimental data processing was performed with the help of the FTIR software Horizon MB version 3.4.0.3 (ABB Inc., Québec, QC, Canada). A CARL ZEISS Axio Imager A1m microscope (Carl Zeiss Microimaging Gmbh, Göttingen, Germany), with polarized, transmitted and reflected light, equipped with digital camera for image acquisition and AxioVision Release 4.8.1 software for image processing, was employed to explore the morphology and to determine the microcrystalline phases based on color differences. For the microscopic study, the samples were embedded in EpoThin–Buehler resin, sanded on abrasive paper and polished onto a Lecloth-type cloth, soaked in a suspension of α-alumina in water and immersed in cedar oil. The illumination source was 100 HAL lamp of the optical microscope. The image was captured with a Canon Power Shot A 640 digital camera (Melville, NY, USA), 4× digital zoom. Morphology and semi-quantitative analysis of iron oxide nanopowders was further investigated using scanning electron microscopy (SEM) coupled with Energy-dispersive X-ray analysis (EDS). SEM/EDS characterization was performed with a Quanta 250 scanning electron microscope (FEI, Eindhoven, Netherlands) of high resolution, fully digitized, and an Energy-dispersive X-Ray spectrometer consisting of ELEMENT Silicon Drift Fixed Detector, and ELEMENT EDS Analysis Software Suite, manufactured by EDAX (Draper, UT, USA). The samples were coated by gold to gain better conductivity required for high quality SEM imaging. Differential Scanning Calorimetry (DSC) was performed using a NETZSCH DSC 200 F3 Maia differential scanning calorimeter (NETZSCH-Gerätebau GmbH, Selb, Germany) in Ar atmosphere worked in Al crucibles up to 590 °C, with a heating rate of 10 K/min and cooling of 30 K/min. The processing of the experimental data was done with the help of the thermal analysis software PROTEUS 7.0 (NETZSCH-Gerätebau GmbH, Selb, Germany).

Differential Scanning Calorimetry coupled with thermogravimetry (DSC-TG) was performed with the Setaram Setsys Evolution apparatus (Setaram Instrumentation, Caluire, France) under inert gas atmosphere (Ar/He), alumina crucibles, with heating rates of 10–50°/min, from room temperature to 1000 °C, then 5 heating-cooling cycles up to 800 °C at 30°/min (for NV5, NV7 samples) and 20°/min respectively (for NV4 and NV6 samples). Thermal Analysis Software Calisto v1.097 (by Setaram Instrumentation, Caluire, France) was used to process the experimental data.

The magnetic measurements have been performed by SQUID (Superconducting Quantum Interference Device) magnetometry. The MPMS 7T (Quantum Design, San Diego, CA, USA) machine was used, as working under the most sensitive Reciprocal Space Option [44]. ZFC-FC (Zero Field Cooled-Field Cooled) measurements under an applied field of 80 Oe (0.008 T magnetic field induction) have been performed in the temperature interval from 10 K to 300 K. This procedure consists of cooling down a system of nanoparticles from the superparamagnetic state to the magnetic frozen state in the absence of any applied magnetic field. The magnetization of the sample is further measured in an applied low magnetic field (e.g. the measuring field of 80 Oe in this case), at increasing temperature up to the superparamagnetic regime. This measured curve is called the ZFC curve because it was obtained after a zero field cooling procedure (see energy implications in [45]). Further on, the sample is again cooled down, this time in an applied magnetic field (usually the measuring field) down to the initial lowest temperature and measured again (e.g., in the same measuring field) at increasing temperature up to the superparamagnetic regime (FC curve). While the FC curve is reversible, it can be measured directly in the same measuring field by decreasing temperature from the superparamagnetic state, just after completing the ZFC curve. In addition, magnetic hysteresis loops have been collected on the investigated samples at 10 K and 300 K. 

The Fe phase composition and local spin structure have been investigated by 57Fe Mössbauer spectroscopy using a constant acceleration spectrometer from SEECo, Edina, MN, USA. Mössbauer spectra at 295 K and 6 K have been acquired by inserting the sample in a close cycle cryostat (JANIS, Woburn, MA, USA). The isomer shifts are reported relative to α-Fe.

## 3. Results and Discussion

### 3.1. X-Ray Diffraction (XRD) Characterization 

The main crystalline phases identified in the iron oxide powders obtained by the hydrothermal process as well as their crystallite size are described in Table 2.

Figure 1 presents the XRD patterns of the investigated samples.

X-ray diffraction highlights the formation of hematite as the major crystalline phase in all samples synthesized by the hydrothermal process. 

The main characteristic peaks of hematite (Fe_2_O_3_) were identified at 2θ = 33.165°, which corresponds to the (104) diffraction plane and at 2θ = 35.608°, corresponding to the (110) diffraction plane. The small peak visible at 2θ = 21.24° is attributed to the goethite structure, FeO(OH) structure corresponding to the (110) plane.

The crystallite sizes presented in Table 2 were calculated using Scherrer formula and the values represent the sizes on the (104)–growth direction (hematite) and (110) growth direction (goethite). It is worth noting that, in the case of NV6 sample, synthesized at a pressure of 100 atm, it is observed that formation of goethite as the secondary phase (~3%), probably due to higher working pressure compared to NV4 samples (20 atm) and NV5 (60 atm). It can be observed that the formation of FeO(OH) structure as a secondary phase is favored by pressure [46,47]. In addition, its crystallite size tends to increase with increasing pressure, while the crystallite size of the Fe_2_O_3_ tends to decrease. 

### 3.2. FT-IR Analysis

FT-IR analysis of iron oxide based powders is depicted in Figure 2.

In the samples NV6 and NV7, the presence of the following vibration bands specific for goethite, and marked with G in Figure 4, was revealed: 905 and 800 cm^−1^ in the NV6 sample, respectively, 897 and 797 cm^−1^ in the NV7 sample. In addition, the stretching vibration of the O-H group at 3123 cm^−1^ in the sample NV6 and 3140 cm^−1^ in the sample NV7 is attributed to goethite, according to Betancur et al. [48]. In all the samples, one can see stretching vibration bands of the OH groups in the range 3600−3200 cm^−1^, located on the surface of the iron oxides [8]. In addition, the peaks corresponding to the deformation vibrations of the OH groups (1653 cm^−1^ for sample NV4, 1693 cm^−1^ for sample NV5, 1668 cm^−1^ for sample NV6 and 1649 cm^−1^ for sample NV7) are revealed. The stretching vibration of the OH group specific to the hematite structure at 3362 cm^−1^ in the NV6 sample confirms the results obtained by XRD analysis according to which this sample is a mixture of hematite and goethite. In addition, the characteristic bands of Goethite in the NV7 sample confirm the hypothesis formulated based on the UV-VIS results, NV7 having a crystalline structure predominantly composed of goethite. The bands in the range 1335–1524 cm^−1^ observed especially in the samples NV4, NV6, and NV7 are probably due to carbonation of the powders, the band at 1335 cm^−1^ being assigned to the (HCO_3_)^−^ group and the band at 1524 cm^−1^ to the group of (CO_3_)^2−^ [49]. Specific Fe-O hematite vibration occurs in NV4 sample at 646 cm^−1^ and NV7 at 644 cm^−1^, respectively. 

### 3.3. Optical Microscopy Characterization (OM)

As one can observe in Figure 3, the samples consist of aggregates made of microcrystalline material, hematite α-Fe_2_O_3_, and possible iron oxy-hydroxides such as goethite α-FeO(OH). Sample NV4 consists mainly of hematite, represented by the red areas in Figure 3a, while the goethite is present in very small quantity, being represented by the gray areas on the edge. In accordance with XRD results, the presence of goethite (gray areas) can be observed in the case of sample NV6 next to white granules of hematite. The white color of hematite is explained by formation of larger-sized particles. The characteristic blood-red color of the hematite is specific to small granules and can be observed in the case of NV7 sample. The more abundant internal reflection is due to both phases (hematite and goethite). Less abundant internal reflexes can be observed in sample NV5. Particle granulation may be higher. Color effects may also be due to anisotropy [50,51,52,53].

### 3.4. SEM-EDS Characterization

The existence of hematite as major phase in hydrothermally synthesized samples is also demonstrated by SEM-EDS characterization, whereas goethite may appear as a secondary phase. An example of phase distribution inside NV4 sample is shown in Figure 4. 

In Figure 4a, goethite phase is represented by some small light gray areas inside hematite.

Figure 4b,d show the distribution of O and Fe content along the line starting from the hematite predominant area (H) to a small goethite area (G). It can be seen that both oxygen and iron content are relatively constant, suggesting that NV4 sample consists of almost 100% hematite, as resulted from XRD characterization.

### 3.5. DSC Analysis 

The results of the differential scanning calorimetry (DSC) analysis are shown schematically in Table 3.

Endothermic peaks in the range of 39.7–100.6 °C are due to water desorption, while endothermic peaks at 148.6 °C (sample NV4) and 241.1 °C (sample NV7) are due to dehydration of the powder. Another endothermic peak observed in sample NV7 at 296 °C could be attributed to decomposition of goethite into hematite. The exothermic peak at 231.2 °C corresponding to the NV6 sample could be explained by the removal of structural water adsorbed physically on the surface of the oxide. Exothermic peaks ranging from 338.9–478.9 °C are probably due to polymorphic transformations from γ-Fe_2_O_3_ (maghemite) into α-Fe_2_O_3_ (hematite). These transformations could take place in several steps, from 292.4 (in the case of the NV5 sample) and 308.7 °C (in the case of the NV4 sample) because of the nanometric dimensions, knowing that the nanoparticles have a different thermal behavior compared to the classical materials. The phase transitions between 290 and 480 °C (supported by exothermic peaks) revealed that the synthesized samples (NV4-NV7) are well crystallized. The results are consistent with particle size analysis by X-ray diffraction (shown in Table 2), confirming the crystallization of the synthesized iron oxides (mainly hematite) [47,54]

### 3.6. Complex Thermal Analysis Characterization (DSC-TG)

The results of the DSC-TG analysis are shown schematically in Table 4.

Exothermic maxima occur in temperature range 250–340 °C (probably due to secondary compounds or unreacted precursors) and 450–550 °C (probably due to polymorphic transformation γ-Fe_2_O_3_ to α-Fe_2_O_3_). In addition to the endothermic maximum due to water loss (57–137 °C), another endothermic effect occurs at 680 °C, regardless of the heating rate, representing the Curie temperature of the hematite [55]. In order to compare the crystalline phases and crystallite sizes of iron oxides before and after thermal treatment at 680 °C, the resulted powders were subjected to morpho-structural characterization.

### 3.7. Morpho-Structural Characterization of Thermally Treated Samples

#### 3.7.1. XRD Characterization

XRD patterns of samples after thermal treatment at 680 °C, denoted as NVx-TT, are depicted in Figure 5, while its corresponding crystallite sizes are presented in Table 5. 

After thermal treatment at 680 °C, the increasing of crystallite size (from 20–40 nm to 80–90 nm) and the partial transformation of hematite into magnetite can be observed.

#### 3.7.2. Optical Microscopy Characterization

Optical microscopy images of the thermally treated samples (NVx-TT) are presented in Figure 6.

The resulted obtained from optical microscopy are in agreement with XRD analysis, showing that, after thermal treatment, samples consisted of: >10% hematite and % magnetite in the case of NV4-TT sample; ~50% hematite and 50% magnetite for NV5-TT sample; >10% hematite in the case of NV6-TT sample; and >10% of % magnetite and % hematite for NV7-TT sample. In the case of NV4-TT sample, intense red reflexes characteristic of hematite can be observed, while the presence of magnetite is suggested by the grey areas. In most granules, magnetite and hematite are intimately associated but are not evenly distributed. The ratio between them is different depending on the area [50,51,52,53]. For NV5-TT sample, predominant areas of hematite (red color) and magnetite (grey color) are clearly delimited.

In sample NV6-TT, the distribution of magnetite on cracks is observed (Figure 6c). Larger hematite granules (white color) are formed in NV7-TT sample (Figure 6d). The presence of magnetite phase is confirmed by the gray areas. The two phases are relatively homogenous and uniformly distributed.

#### 3.7.3. SEM-EDS Characterization

The existence of both hematite and magnetite phases in thermally treated samples is also demonstrated by SEM-EDS characterization. An example of phase distribution inside NV4-TT sample is shown in Figure 7. In Figure 7a, magnetite phase is represented by the small white areas and the cracks formed due to hematite reduction. Figure 7b,d clearly show the distribution of O and Fe content along the line starting from the hematite predominant area (M) to magnetite predominant area (H). It can be seen that the oxygen content decreases with the transition from H (30% O) to M (27.6%), as expected.

### 3.8. Magnetic Characterization

ZFC-FC curves and hysteresis loops at 10 K and 300 K are shown in Figure 8 and Figure 9 for samples NV4 and NV4-TT. As a direct observation related to sample NV4 is the step-like increment of the magnetization in the ZFC curve at about 250 K and the permanently higher magnetization in the FC curve as compared to the ZFC one. The first aspect is related to the well-known Morin transition tacking place in the corundum structure of hematite, pointing to a change in the spin structure from antiferromagnetic (at low temperature) to a weak ferromagnetic one at room temperature [56]. Note that such a transition assumes a well crystallized hematite with a nanoparticle size ranging out from the superparamagnetic regime. However, the upper variation of the FC curve over the ZFC one supports the idea of monodomain like nanoparticles, in agreement with sizes of tens of nanometers as obtained from XRD. The change from an antiferromagnetic spin structure at low temperature to a weak ferromagnetic one at room temperature is related to the appearance of a slight canting along the [111] trigonal direction of the spins initially antiferomagnetic coupled in the (111) plane at low temperature [56]. The main antiferromagnetic behavior is also sustained by the lack of saturation with continuous (linear) increment of the magnetization in higher fields as well as by the very low values of magnetization, whereas the above-mentioned change in the spin structure is sustained by the higher magnetization in the loop at 300 K as compared to the loop at 10 K (see Figure 8a,b).

Note the drastically changed ZFC-FC curves and hysteresis loops of sample NV4-TT as compared to the case of sample NV4 (see Figure 8c,d). First of all, the branching aspect with continuous increase of the ZFC curve vs. temperature and continuous increase of FC magnetization at decreasing temperature stand for magnetic mono-domain nanoparticles with blocking temperatures well above 300 K. The small jumps on both ZFC and FC curves evidenced in the range of 100–120 K and 230–250 K suggest the formation of poorly crystallized magnetite phases (with Verwey temperature of about 120 K) and a rest of hematite nanoparticles (with a Morin temperature of about 250 K) [56,57]. This aspect is also definitely supported by the value of saturation magnetization, e.g., about 28 emu/g (3.51 × 10^−5^ Tm^3^/kg) at 10 K as resulting from the loops presented in Figure 8d. Assuming that a well crystallized magnetite might have some 90 emu/g (11 × 10^−5^ Tm^3^/kg) saturation magnetization [45,57], it results in a rough amount of a maximum of 28% of magnetite in the sample. In the inset of the same figure, the loops over a narrower field range are presented, pointing for the presence of two magnetic phases, the one giving the dominant magnetic signal having a coercive field of about 450 Oe (0.045 T magnetic field induction) at 10 K and 190 Oe (0.019 T magnetic field induction) at 300 K, specific to magnetite.

The magnetic measurements obtained on samples NV6 and NV6-TT are shown in Figure 9. The first aspect derived from ZFC-FC measurement is the lack of any trace of nanoparticle related dynamical behavior of superspins in the NV6 system (i.e., no increment of the ZFC curve at low temperatures). 

This suggests that, even in a case of nanoparticle-like morphology, these interact magnetically not only by weak dipolar interactions, but they are much stronger. 

On the other hand, the Morin transition is very clearly evidenced in this sample together with its interesting temperature loop (Figure 9a,b). It is to be concluded that well-formed and strongly interacting hematite particles are found, giving rise to an overall structure of magnetic domains at 300 K, as evidenced via the corresponding hysteresis loop in Figure 9b.

Note the same characteristics of the hysteresis loops for this sample as for sample NV4, except a much larger coercive field at 300 K, namely 500 Oe (0.05 T magnetic field induction) as compared to 140 Oe (0.014 T magnetic field induction), in case of negligible coercive fields at 10 K for both samples.

The thermal annealing changes again the phase composition as clearly evidenced from Figure 9c, where again two magnetic transitions can be deduced from both ZFC and FC curves (the Verwey transition at 100–120 K and Morin transition at 220–260 K). Both transitions are much better evidenced as compared to sample NV4-TT (see Figure 9c) in conditions of a much lower magnetization at saturation of only 15 emu/g (1.88 × 10^−5^ Tm^3^/kg) at 10 K (see Figure 9d). This means that the amount of magnetite is lower in sample NV6-TT (e.g. less than 17%), but it is of much better quality (e.g., better crystallized) than in the case of sample NV4-TT. This fact is supported also by the enhanced coercive fields in this sample, e.g., of 840 Oe (0.084 T magnetic field induction) at 10 K and 320 Oe (0.032 T magnetic field induction) at 300 K. 

Finally, the magnetic results on samples NV7 and NV-7TT are shown in Figure 10. 

The first aspect to be mentioned is the higher magnetization and the trend of the ZFC curve of NV7 sample (Figure 10a) with a maximum at a temperature of about 50 K specific to nanoparticulate systems with enhanced magnetic moments (e.g., no antiferromagnetic coupling) and a blocking temperature of 50 K. However, the type of branching supports also the presence of an additional magnetic phase which intermediates a sort of inter-particle interactions. The magnetization values in high fields are, according to Figure 10b, an order of magnitude higher than in the case of the previous samples, giving support for a magnetic phase different from hematite. This additional phase that is magnetic at low temperatures and with paramagnetic/superparamagnetic behavior at 300 K can be assigned to a Fe oxo-hydroxide, whose nature might be better determined by Mössbauer spectroscopy.

Finally, following the thermal treatment, the specific magnetite and hematite phases are again evidenced through the specific Verwey and Morin transitions evidenced by the ZFC-FC measurements on sample NV7-TT (Figure 10c). The saturation magnetization is close to 30 emu/g (3.77 × 10^−5^ Tm^3^/kg) at both 10 K and 300 K (Figure 10d), proving quite large magnetite particle sizes). Together with the well pronounced jump along the Verwey transition, this result supports the formation of well crystallized and relatively large magnetite nanoparticles with a relative concentration close to 33 wt%. 

Mössbauer spectra of samples NV4 and NV4-TT (at 6 K and 300 K for each sample) are shown in Figure 11 (a and b for NV4; c and d for NV4-TT). Mössbauer spectra of samples NV5, NV6, and NV7 (at 10 K and 300 K for each sample) are shown in Figure 12 (a, b for NV5; c, d for NV6; and e, f for NV7). It can be observed that the as prepared sample NV4 consists at 6 K and 300 K in only one sextet component with hyperfine parameters specific to hematite (e.g., hyperfine magnetic field of 54 T at 6 K and 51 T at 300 K). After the thermal treatment, an additional phase fitted by one Mössbauer sextet at 6 K and two Mossbauer sextets at 300 K are observed. The hyperfine parameters of this new phase are specific to magnetite (e.g., at 300 K the hyperfine magnetic fields are of 49.2 T and 46.3 T). The magnetite represents 24% from the Fe phases, with the other 76% remaining as hematite. Note the very defect structure of magnetite with an occupation of 1:1 on octahedral to tetrahedral positions.

Concerning the behavior of the rest of as prepared samples, NV5 is also represented by only one component with specific hyperfine parameters for hematite, NV6 presents an additional component (3 wt%) with hyperfine parameters specific to goethite (e.g., 50.3 T at 6 K and 38.2 T at 300 K), in excellent agreement with the XRD characterization and, finally, sample NV7 is formed by 56% of hematite and 44% of goethite, according to the low temperature Mössbauer spectrum (note that MS is more sensitive to the composition of the Fe phases as compared to XRD). At RT, about 20% of goethite becomes superparamagnetic within the Mössbauer time window, again in excellent agreement with the magnetic measurements pointing for superparamagnetic behavior of goethite nanoparticles at higher temperatures. 

The as evidenced type of involved magnetic phases and their distribution in the analyzed samples are shown in Table 6 and Figure 13.

## 4. Conclusions

The influence of synthesis pressure on crystalline structure of iron oxide prepared by hydrothermal method was studied. It has been found that, for lower pressure values (less than 100 bar), iron oxide is predominantly formed as hematite, while, at pressures above 100 bar, the major crystalline phase is goethite. The complex thermal analysis by the DSC method revealed the polymorphic changes of iron oxides at different temperatures. Thermal stability of hydrothermal synthesized nanoparticles under various experimental conditions has been demonstrated by performing heating/cooling cycles. 

The specific magnetite and hematite phases have been evidenced in all thermally treated samples through XRD, optical microscopy, SEM characterization, and the specific Verwey and Morin transitions evidenced by the ZFC-FC measurements. ZFC and FC curves suggested the formation of poorly crystallized magnetite (maximum 28%) phases (with Verwey temperature of about 120 K) and a rest of hematite nanoparticles (with a Morin temperature of about 250 K) in the case of NV4-TT sample. The amount of magnetite is lower in sample NV6-TT (e.g., less than 17%), but it is much better crystallized than in the case of sample NV4-TT. The formation of well crystallized and relatively large magnetite nanoparticles with a relative concentration close to 33 wt% was observed in the case of NV7-TT sample. The results obtained by Mössbauer characterization are in excellent agreement with the XRD and MO characterization, showing the formation of one component in the case of samples prepared at lower pressure (<100 bar)–hematite (with partial transformation in magnetite after thermal treatment), and the appearance of goethite for samples synthesized at pressures >100 bar. It has been proven that the phase composition and magnetic parameters of the samples can be tuned via the applied pressure during the hydrothermal synthesis as well as by subsequent annealing conditions. Among the as prepared samples, of very weak ferromagnetic contributions, one order of magnitude higher magnetization is obtained for sample NVT synthesized under the highest applied pressure (e.g. 1000 bar). The same sample also provides, by a subsequent annealing treatment, the highest amount of good quality magnetite leading to a magnetization at saturation of 30 emu/g (3.77 × 10^−5^ Tm^3^/kg) and a coercive field of 1000 Oe (0.1 T magnetic field induction) at 10 K and 450 Oe (0.045 T magnetic field induction) at 300 K, already convenient for various applications.

## Figures and Tables

**Figure 1 nanomaterials-10-00085-f001:**
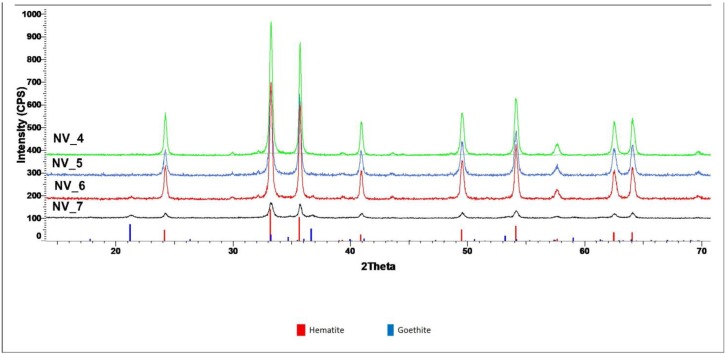
X-ray diffraction (XRD) patterns of samples NV4, NV5, NV6, and NV7, compared to the XRD patterns of hematite (red) and goethite (blue) from the International Centre for Diffraction Data Powder Diffraction File (ICDD PDF) 4+ 2019 data base.

**Figure 2 nanomaterials-10-00085-f002:**
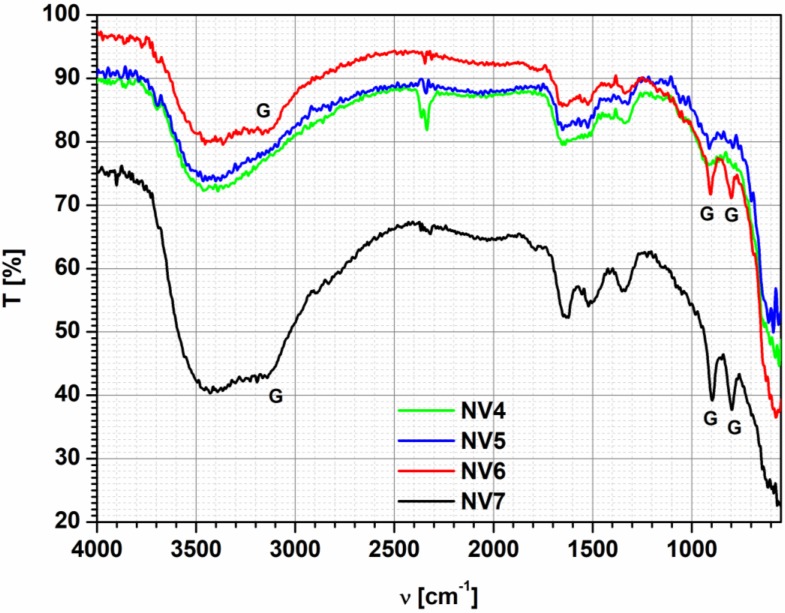
Fourier transform infrared (FT-IR) spectra of iron oxide samples synthesized in hydrothermal conditions at high pressure: NV4 (20 bar), NV5 (60 bar), NV6 (100 bar), and NV7 (1000 bar).

**Figure 3 nanomaterials-10-00085-f003:**
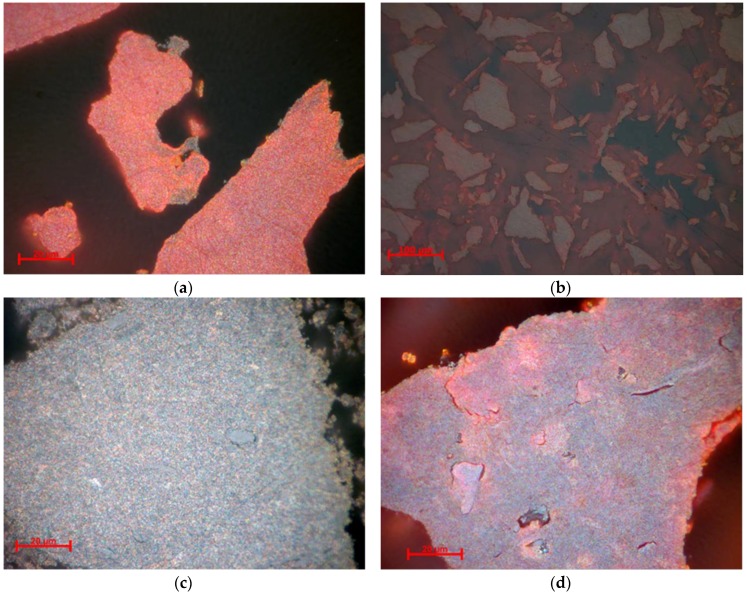
Reflected light, crossed Nicols, immersion in cedar oil. Microcrystalline aggregates in the sample: (**a**) NV4-Hematite (white, red internal reflections), magnetite (gray); (**b**) NV5-Hematite (white, red internal reflections), magnetite (gray); (**c**) NV6-Magnetite (gray), hematite (white, red internal reflections); (**d**) NV7-Hematite (white, red internal reflections), magnetite (gray).

**Figure 4 nanomaterials-10-00085-f004:**
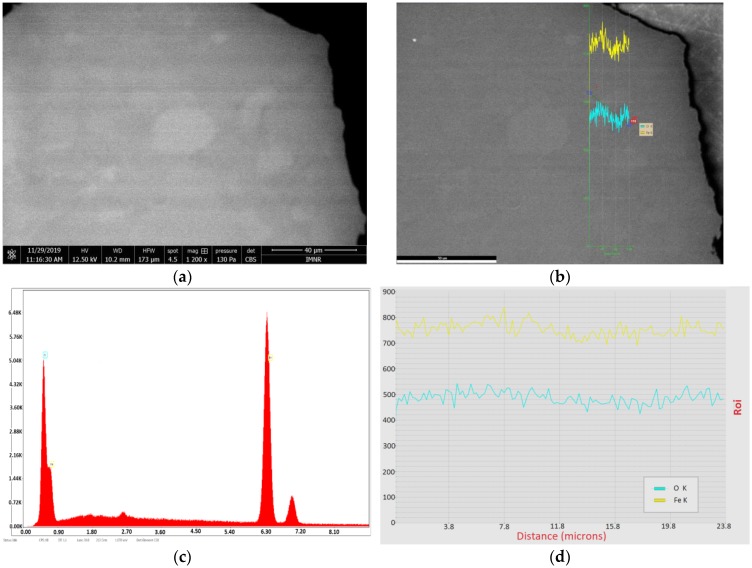
Scanning Electron Microscopy with Energy Dispersive Spectroscopy (SEM-EDS) characterization of a hematite sample (NV4) prepared by hydrothermal synthesis at 20 bar, embedded in EpoThin-Buehler resin: (**a**) micrograph by scanning electron microscopy (SEM)collected using Circular Backscatter Detector (CBS); magnification 1200×; scale bar 40 µm; (**b**) line between hematite and goethite areas; (**c**) elemental analysis by energy dispersive X-ray spectroscopy (EDS); (**d**) distribution of O (turquoise) and Fe (yellow) content along the line between hematite and goethite areas from left to right.

**Figure 5 nanomaterials-10-00085-f005:**
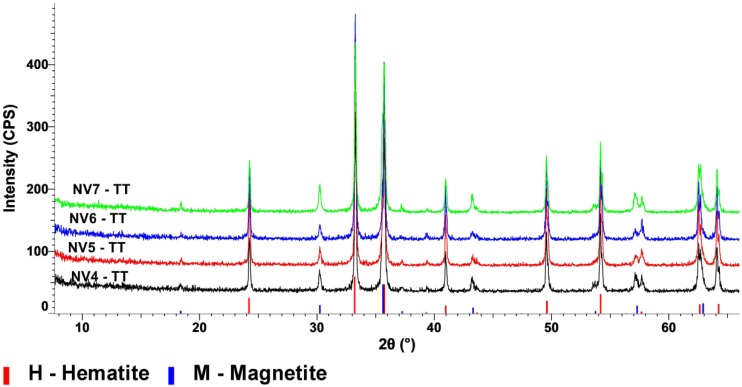
X-ray diffraction (XRD) patterns of thermally treated samples NV4-TT, NV5-TT, NV6-TT, and NV7-TT, compared to the XRD patterns of hematite (red) and magnetite (blue) from the ICDD PDF 4+ 2019 data base.

**Figure 6 nanomaterials-10-00085-f006:**
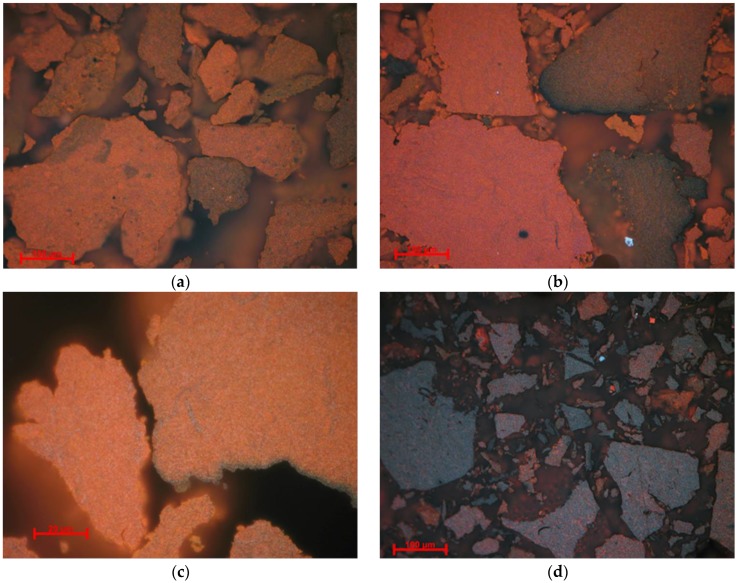
Reflected light, crossed Nicols. Optical micrographs of the thermally treated samples: (**a**) NV4-TT; (**b**) NV5-TT, Hematite (white, red internal reflections), magnetite (gray); (**c**) NV6-TT, Hematite (white, red internal reflections), magnetite (gray); (**d**) NV7-TT, Magnetite (gray), hematite (white, red internal reflections).

**Figure 7 nanomaterials-10-00085-f007:**
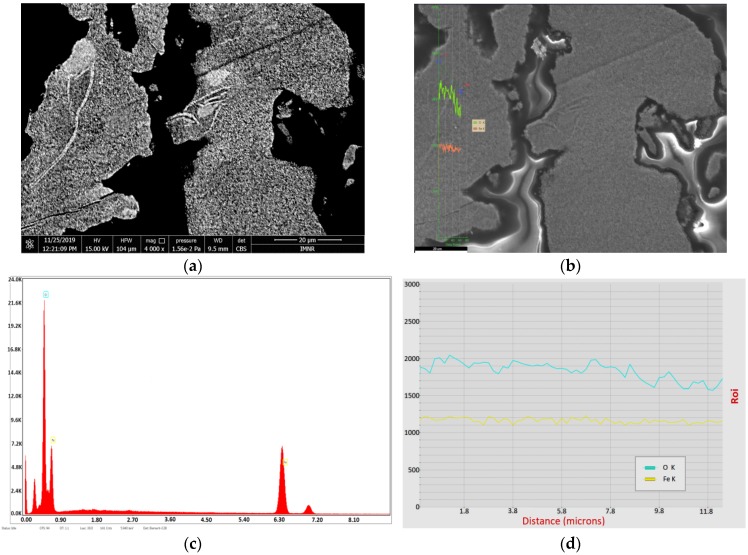
SEM-EDS characterization of a hematite sample after thermal treatment at 680 °C (NV4-TT), embedded in EpoThin-Buehler resin: (**a**) micrograph by scanning electron microscopy (SEM) collected using Circular Backscatter Detector (CBS); magnification 4000×; scale bar 20 µm; (**b**) line between hematite and magnetite areas; (**c**) elemental analysis by energy dispersive X-ray spectroscopy (EDS); (**d**) distribution of O (turquoise) and Fe (yellow) content along the line between hematite and magnetite areas from left to right.

**Figure 8 nanomaterials-10-00085-f008:**
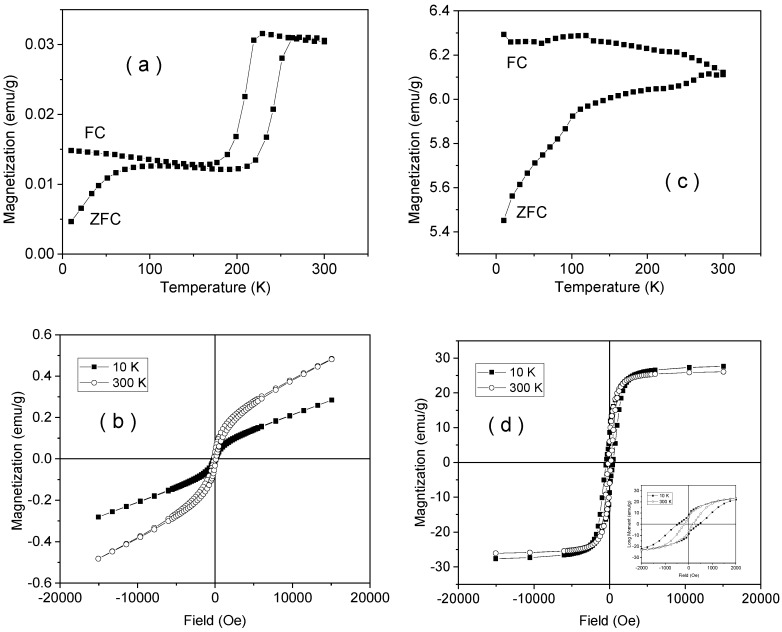
Zero Field Cooled-Field Cooled (ZFC-FC) curves collected in 80 Oe (0.008 T magnetic field induction) direct current (DC) applied field (**a**) and hysteresis loops (**b**) for sample NV4, similar ZFC-FC curve (**c**) and hysteresis loops (**d**) for sample NV4-TT.

**Figure 9 nanomaterials-10-00085-f009:**
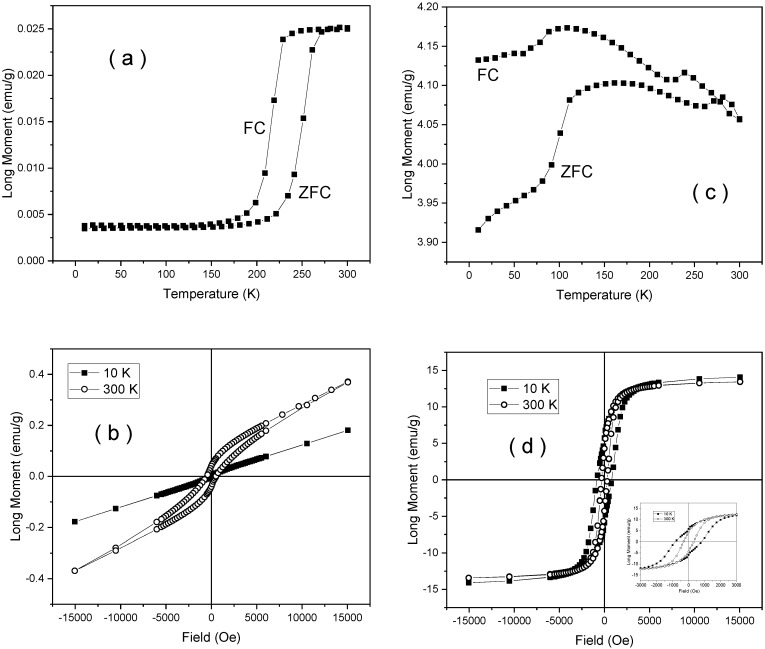
ZFC-FC curves collected in 80 Oe (0.008 T magnetic field induction) DC applied field (**a**) and hysteresis loops (**b**) for sample NV6, similar ZFC-FC curve (**c**) and hysteresis loops (**d**) for sample NV6-TT.

**Figure 10 nanomaterials-10-00085-f010:**
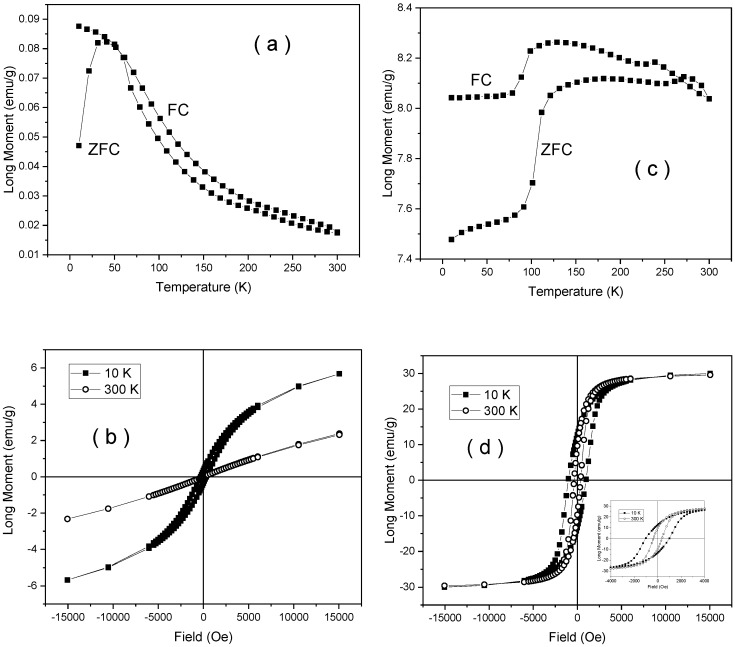
ZFC-FC curves collected in 80 Oe (0.008 T magnetic field induction) DC applied field (**a**) and hysteresis loops (**b**) for sample NV7, similar ZFC-FC curve (**c**) and hysteresis loops (**d**) for sample NV7-TT.

**Figure 11 nanomaterials-10-00085-f011:**
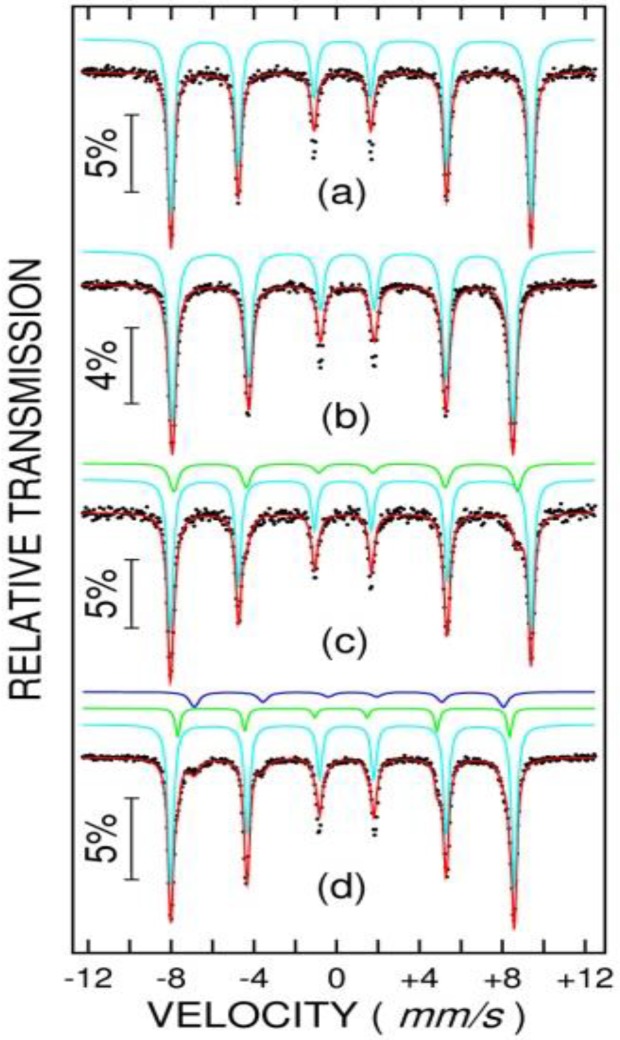
Mössbauer spectra of sample NV4 at 6 K and 300 K (**a**,**b**) and NV4-TT (**c**,**d**). Black dots represent the experimental data and the red curve is the theoretical fit as obtained by superposing the spectral components represented above each spectrum.

**Figure 12 nanomaterials-10-00085-f012:**
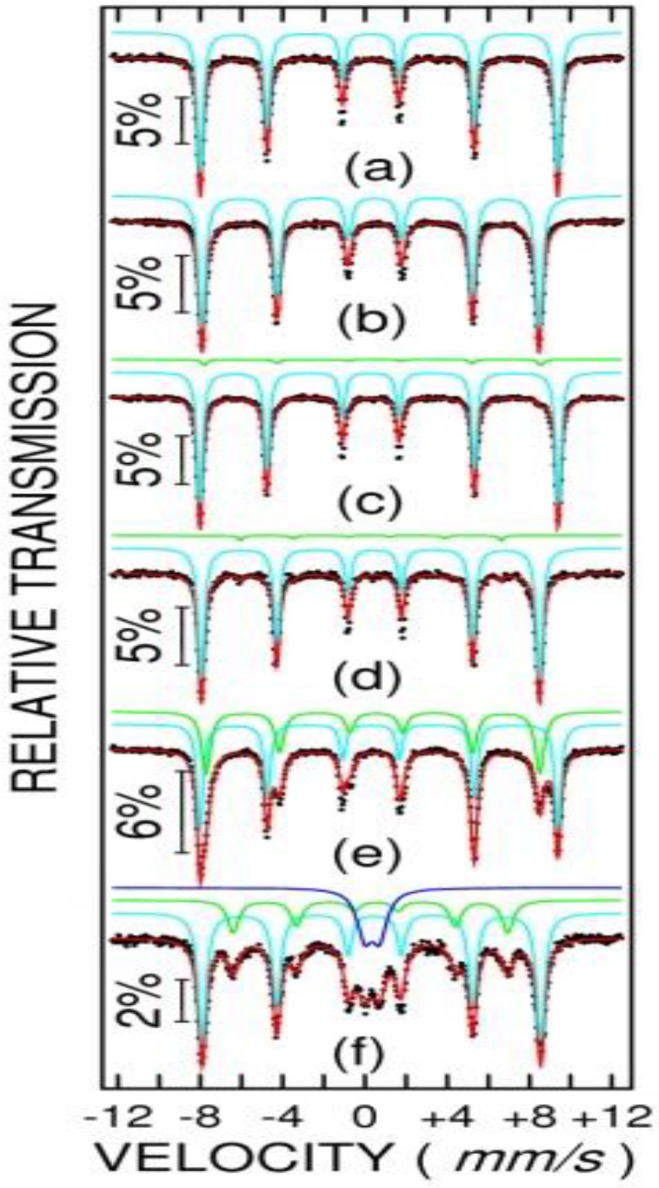
Mössbauer spectra of sample NV5 at 6 K and 300 K (**a**,**b**), NV6 (**c**,**d**) and NV7 (**e**,**f**). Black dots represent the experimental data and the red curve is the theoretical fit as obtained by superposing the spectral components represented above each spectrum.

**Figure 13 nanomaterials-10-00085-f013:**
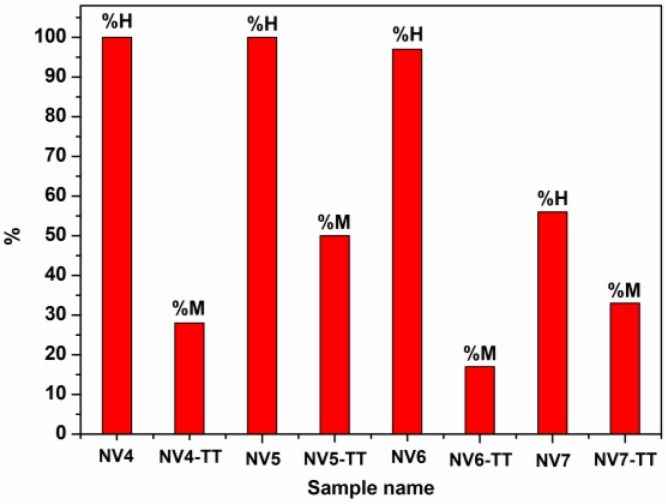
Distribution of hematite (H) and magnetite (M) in all iron oxide samples synthesized by hydrothermal method in high pressure conditions, before (NVx) and after thermal treatment at 680 °C (NVx-TT).

**Table 1 nanomaterials-10-00085-t001:** Synthesis conditions of iron oxide nanostructures prepared in this work and Fe content of the investigated samples, according to quantitative chemical analysis.

Sample Name	Synthesis Conditions	Material Structure	Fe, wt%
NV4	200 °C/3 h/20 bar	hematite	68.87
NV5	200 °C/3 h/60 bar	hematite	68.41
NV6	200 °C/3 h/100 bar	hematite	68.32
NV7	100 °C/3 h/1000 bar	hematite/goethite	61.74

**Table 2 nanomaterials-10-00085-t002:** Crystallite size (calculated using Scherrer formula) and crystalline phases of hydrothermally synthesized iron oxide nanostructures (NVx), determined by X-ray diffraction (XRD), according to the reference patterns in the International Centre for Diffraction Data Powder Diffraction File (ICDD PDF) 4+ 2019 database.

Sample	Crystallite Size (Scherrer), nm	The Crystalline Phase Identified by XRD; Phase Content *	Formula	PDF References
NV4	36	100% Hematite	Fe_2_O_3_	01-089-0599
NV5	32	100% Hematite	Fe_2_O_3_	01-089-0599
NV6	37 22	96.7% Hematite 3.3% Goethite	Fe_2_O_3_ FeO(OH)	01-089-0599 01-081-0464 (I)
NV7	33 25	~66% Hematite ~34% Goethite	Fe_2_O_3_ FeO(OH)	01-077-9924 (*) 00-029-0713 (I)

* It should be noted that the percentage of crystalline phases calculated from XRD is a semi-quantitative estimation.

**Table 3 nanomaterials-10-00085-t003:** Thermal analysis results obtained by Differential Scanning Calorimetry (DSC) for iron oxide nanostructures (NVx) synthesized by hydrothermal process. Main characteristics of endothermic and exothermic peaks (temperature and enthalpy values) observed from room temperature (RT) up to 590 °C.

Sample	Peak 1		Peak 2		Peak 3		Peak 4		Peak 5	
	T, °C	ΔH, J/g	T, °C	ΔH, J/g	T, °C	ΔH, J/g	T, °C	ΔH, J/g	T, °C	ΔH, J/g
NV4	39.7	2.4	148.6	0.97	308.7	−10.7			478.9	−4.56
NV5	40.7	2.05	67.1	1.6	292.4	−0.88	366.2	−1.13	441.8	−1.33
NV6	84.5	1.65			231.2	−0.64	338.9	−3.32		
NV7	100.6	82.86	241.1	9.06	296	5.21			413.1	−21.47

**Table 4 nanomaterials-10-00085-t004:** Results obtained by simultaneous thermal analysis: differential scanning calorimetry coupled with thermogravimetry (DSC-TG) for iron oxide nanostructures (NVx) synthesized by hydrothermal process. Five heating-cooling cycles have been performed for each sample up to 800 °C.

Sample	Speed	Peak 1		Peak 2		Peak 3		Peak 4		Peak 5	
	°/min	T, °C	ΔH, J/g	T, °C	ΔH, J/g	T, °C	ΔH, J/g	T, °C	ΔH, J/g	T, °C	ΔH, J/g
NV4 5 cycles with 20°/min up to 800°C	10	57.04	5.96	291.44	−9.19 exo					676.29	0.30
	Δm, %	−0.476		−0.222						−0.041	−1.578
	20			258.56	−1.63 exo			516.69	−2.07 exo	678.41	0.33
	Δm, %			−0.131				−0.027		−0.044	−1.794
	30	91.38	5.86	320.72	−2.73 exo	389.34	−0.13 exo	449.94	0.39	677.92	0.30
	Δm, %	−0.59		−0.168		−0.035		-0.053		−0.036	−1.686
	50	84.29	0.27	339.48	0.28					679.23	0.38
	Δm, %	−0.167		−0.101						−0.018	−1.737
NV5 5 cycles with 30°/min up to 800 °C	10	70.79	7.10	257.45	0.72					680.94	0.26
	Δm, %	−0.395		−0.152						−6.25 × 10^−3^	−1.556
	20	92.83	5.56	299.67	−5.22 exo	380.17	−0.55 exo	466.56	−0.56 exo	678.68	0.39
	Δm, %	−0.537		−0.255		−0.057		−0.041		−0.043	−1.546
	30	106.47	21.87	273.97	−1.08 exo	370.96	−8.15 exo	512.55	−1.51 exo	679.26	0.36
	Δm, %	−0.803		−0.095		−0.32		−0.126		−0.039	−1.751
	50	121.51	17.05							680.32	0.58
	Δm, %	−0.686								−0.018	−1.501
NV6 5 cycles with 20°/min up to 800 °C	10	67.72	4.35	278.29	−7.76 exo			479.11	−6.27 exo	679.00	0.38
	Δm, %	−0.368		−0.49				−0.121		−0.028	−1.513
	20	94.50	4.73			375.16	−14.00 exo	499.98	−2.13 exo	679.23	0.74
	Δm, %	−0.402				-0.573		−0.1		−0.04	−1.963
	30	104.36	6.26	247.83	−0.93 exo	390.41	−5.13 exo			679.83	0.82
	Δm, %	−0.439		−0.156		−0.629				−0.012	−1.518
	50	122.74	19.63			411.07	−3.92 exo	557.59	−2.26 exo	678.89	1.93
	Δm, %	−0.678				−0.559		−0.032		-0.072	−1.562
NV7 5 cycles with 30°/min up to 800 °C	10	89.29	84.66	269.75	−6.93 exo					680.95	0.26
	Δm, %	−5.821		−1.604						−7.88 × 10^−3^	−9.425
	20	120.50 184.86	91.87 6.46	297.43	−5.83 exo			469.56	−1.19 exo	679.09	0.43
	Δm, %	−5.865 −0.718		−1.462				−0.178		-0.065	−10.27
	30	118.62	96.00	290.07	−4.79 exo					680.49	0.51
	Δm, %	−6.911		−1.591						-0.018	−10.39
	50	137.24	100.21	302.59	−4.44 exo					679.99	0.61
	Δm, %	−7.424		−1.744						−8.76 × 10^−3^	−10.88

**Table 5 nanomaterials-10-00085-t005:** Crystallite size (calculated using Scherrer formula) and crystalline phases of thermally treated samples (NVx-TT), determined by X-ray difraction (XRD), according to the reference patterns in the ICDD PDF 4+ 2019 database.

Sample Name	Pressure, Bar	Crystallite Size (Scherrer), nm	The Crystalline Phase Identified by XRD; Phase Content *	Formula	PDF References
NV4-TT	20	81.6	77% Hematite 23% Magnetite	Fe_2_O_3_ Fe_3_O_4_	PDF 04-003-5818 (P) PDF 04-007-2718 (*)
NV5-TT	60	93	86% Hematite 14% Magnetite	Fe_2_O_3_ Fe_3_O_4_	PDF 04-003-5818 (P) PDF 04-007-2718 (*)
NV6-TT	100	82.6	89% Hematite 11% Magnetite	Fe_2_O_3_ Fe_3_O_4_	PDF 04-003-5818 (P) PDF 04-007-2718 (*)
NV7-TT	1000	86.4	72% Magnetite 28% Hematite	Fe_3_O_4_ Fe_2_O_3_	PDF 04-007-2718 (*) PDF 04-003-5818 (P)

* It should be noted that the percentage of crystalline phases calculated from XRD is a semi-quantitative estimation.

**Table 6 nanomaterials-10-00085-t006:** Phase distribution in all samples.

Sample Name	Phase Type
NV4	hematite
NV4-TT	Hematite + magnetite
NV5	hematite
NV5-TT	Hematite + magnetite
NV6	Hematite + goethite
NV6-TT	Hematite + magnetite
NV7	Hematite + goethite
NV7-TT	Hematite + magnetite

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
