# Peer review of "The Influence of Synthesis Parameters on Structural and Magnetic Properties of Iron Oxide Nanomaterials"

_nanomaterials, 2020, doi:10.3390/nano10010085_

Round 1
Reviewer 1 Report
The authors investigated the effect of pressure and temperature on the structural, thermal and magnetic properties of iron oxide nanomaterials prepared by hydrothermal synthesis. The referee feels there are several comments as described below:
1) Line 123: 14°÷84° -> 14°-84°
2) Figure 4 and 7: Numerical values ​​on the vertical and horizontal axes are too small to read.
3) Figure 11 and 12: Do black dots mean experimental data? What are some solid lines? The authors should explain details.
4) Figure 13: Enter a numerical value on the vertical axis
Reviewer 2 Report
Referee's report on the paper “nanomaterials-671715: ”The influence of synthesis parameters on structural and c properties of iron oxide nanomaterials,” by M. Cursaru et al.
The authors discuss a comprehensive investigation on the structural and magnetic properties of nanostructured iron oxide powders, with specific attention devoted to the role of pressure applied during the adopted hydrothermal synthesis process. Hematite and goethite are prevalently formed at low and high (more than 100 bar) pressures, respectively. A thermal treatment at 680 °C leads to an increase of the nanoparticle size and to the formation of magnetite. A full array of characterization methods has been adopted: XRD, DSC, optical and electron microscopy, Mössbauer, magnetic measurements. This resulted in a fairly complete assessment of the physical properties of the materials. These properties are discussed to a satisfactory extent and the complex evolution of the magnetic properties as a function of the synthesis pressure, the thermal treatment, and the temperature is interpreted coherently in terms of phase transitions (i.e., Morin and Verwey, antiferromagnetic-magnetic, etc.). The paper is well organized and informative. I believe it will be useful to people working in this field.
Some minor comments:
1) Field-cooled samples. How high the field?
2) Page 11, sub-section b: what do you mean by tens of %? Can you place a defined figure in Table5?
3) I find the captions of figure and tables quite synthetic. I would recommend to provide more details.
4) Why choosing a field of 80 Oe for the measurement of the magnetization versus temperature? You measure in this way a generically small value of the magnetization, which can be influenced by the specific microstructural state of the material, thereby lacking intrinsic features.
5) The SI system for the magnetic quantities should be applied.
Reviewer 3 Report
This is a good paper that gives an analysis of the effect of pressure on the thermal treatment of iron oxide nanoparticles. The methods are sound, and the conclusions are supported by the data. The paper could be improved by the following revisions.
The first sentence has a problem: It should say that the particles have been used for the past 40 years (past tense instead of present tense).
Line 39: particles’ stability in water with an apostrophe.
Line 47: Néel
Line 53+: numerous studies should have more than one reference, please provide all the references here
Line 104: was the mixture washed (with water?)
Line 138: What was the illumination source? This is important for assessing the observed colors.
Line 154+: Magnetometry description has a few problems. First, the reciprocal space option is not well known, so it is suggested that a reference be provided, for example[1]. Also ZFC-FC refer to zero field cooled and field cooled but the description states a field of 80 Oe. This is a pretty small field compared to the hysteresis loop fields. Is ZFC obtained at zero field? This is not clear.
Line 252: Are the different observed temperatures consistent with particle size analysis by xrd?
Line 275: transformation
Line 323: weak
Line 312+: even though the magnetic structures are presumably well-known, references are needed here that give Morin transition temperature, the nature of the spin reorientations, magnetizations, Verway transition, etc.
Figures 8, 9 and 10 should be labeled (a), (b), (c), and (d). The way these figures are labeled with two (a)s and two (b)s is confusing.
Figure 13 needs tick labels. The values of percentages should be shown.
Line 440: How much increase is observed. A quantitative analysis should have a quantitative conclusion. Also, it would be helpful if more quantitative information was included in the abstract, for example, how much improvement is obtained for the best case.
1 M. Buchner, K. Höfler, B. Henne, V. Ney, and A. Ney, "Tutorial: Basic principles, limits of detection, and pitfalls of highly sensitive SQUID magnetometry for nanomagnetism and spintronics," J Appl Phys 124 (16), 161101 (2018).
Round 2
Reviewer 1 Report
This manuscript was properly revised according to the reviewer's comments. It can be accepted as it is.
Reviewer 2 Report
The authors have extensively and satisfactorily replied to the comments and appropriately modified their manuscript, I do not have any further comments to make.
Reviewer 3 Report
The authors have done a good job addressing the criticisms, and the paper is very good.